# Models of Delivery of Sustainable Public Transportation Services in Metropolitan Areas–Comparison of Conventional, Battery Powered and Hydrogen Fuel-Cell Drives

**Tomasz Wojciech Szulc** [1,*], **Grzegorz Krawczyk** [2] **and Seweryn Tchórzewski** [3]

1   Department of Logistics, Silesian University of Technology, 41-800 Zabrze, Poland
2   Department of Transport, University of Economics in Katowice, 40-287 Katowice, Poland; grzegorz.krawczyk@ue.katowice.pl
3   Department of Management, Silesian University of Technology, 41-800 Zabrze, Poland; seweryn.tchorzewski@polsl.pl
*   Correspondence: tomasz.szulc@polsl.pl

**Abstract:** The development of public transport systems is related to the implementation of modern and low-carbon vehicles. Over the last several years, there has been a clear progress in this field. The number of electric buses has increased, and the first solutions in the area of hydrogen fuel cells have been implemented. Unfortunately, the implementation of these technologies is connected with significant financial expenditure. The goal of the article is the analysis of effectiveness of financial investment, consisting in the purchase of 30 new public transport buses (together with the necessary infrastructure–charging stations). The analysis has been performed using the NPV method for the period of 10 years. Discount rate was determined on 4%, as recommended by the European Commission for this type of project. It is based on the case study of the investment project carried out by Metropolis GZM in Poland. The article determines and compares the efficiency ratios for three investment options-purchase of diesel-powered, battery-powered, and hydrogen fuel-cell electric vehicles. The results of the analysis indicate that the currently high costs of vehicle purchase and charging infrastructure are a significant barrier for the implementation of battery-powered and hydrogen fuel-cell buses. In order to meet the transport policy goals related to the exchange of traditional bus stock to more eco-friendly vehicles, it is necessary to involve public funds for the purpose of financing the investment activities.

**Keywords:** effectiveness of transport invest; public transport; financing of charging infrastructure

## 1. Introduction

### 1.1. Public Transport Funding

Public transport constitutes an important element of a city's functioning. It is one of the tools for sustaining urban mobility as an alternative to car transport, especially in home-work-home travel. It also has important social functions, enabling comfortable and relatively cheap movement of inhabitants, including the elderly, the disabled, and low-income individuals [1]. Research shows that in the case of elderly people who have lost the possibility or will to drive a car themselves, public transport is the only opportunity to maintain an active lifestyle and social relationships [2–6]. For the disabled who cannot drive a car or use bicycles, public transport also provides the possibility to meet their transport needs related to work, treatment, and free time. Public transport also provides an opportunity to move for low-income individuals and the unemployed. Thanks to the network of transport connections, such people may extend the territorial scope of their job search and have access to educational services. The above-indicated social aspect is the reason why public transport system cannot only be based on connections characterised by the highest demand during the morning and afternoon peak hours. The public transport offered should also:

- be available in non-peak hours and on non-working days;
- provide the possibility to access places of education, health care units, commercial outlets, marketplaces, offices, and religious sites;
- be carried out by means of modern and user-friendly rolling stock;
- be developed and modified along with the spatial development of the operated areas.

From the social point of view, public transport should not only address the demands of temporal and spatial availability, but also economic accessibility. As a consequence, the tariff policy conducted by local and regional authorities is based on relatively low travel payments (in particular in the case of season tickets), a concessionary fares scheme for different social groups (e.g., students, elderly people, disabled), and, in certain cases, fare-free public transport [7–9]. The revenues from the sale of tickets are insufficient to cover the costs of public transport system operation. Currently, in European conditions, this level amounts to less than 50%, and, in some cases, it reaches only 30%. This means the emergence of deficit needs to be covered by the public. Additionally, public transport systems face numerous challenges, which will have a negative impact on the financial result of the system in the future (see Table 1).

**Table 1.** Factors impacting the level of public transport financing (in Polish conditions).

| Factors Affecting Public Transport System | Expected Changes | Impact on Financing |
|---|---|---|
| Society ageing | Increased number of individuals paying concessionary fares | Decreased revenue from sale of tickets |
| | Decreased number of passengers travelling regularly and fully charged for travel Increased demand for travel between peak hours | Increased volume of operation work affecting the increase of operating costs |
| | Necessity to adjust rolling stock and infrastructure to the needs of elderly people | Increased investment expenditure for rolling stock and infrastructure |
| Depopulation (in the case of certain EU states) | Decreased number of passengers | Decreased demand for public transport travel-decreased revenue from the sale of tickets |
| Suburbanisation | Dispersion of buildings and reduction of average population density in the suburbs. | Increase of operating costs resulting from the operation in new housing with low demand. |
| Implementation and maintenance of new technologies in the field of transport, passenger information and analysis of passenger flows | Ongoing increase of the costs of maintaining modern technologies resulting from the necessity to develop IT infrastructure, update software, and archive data. | Increased costs on the part of the public transport organiser. |
| Replacement of public transport rolling stock with more ecological vehicles | One of the strategic objectives of public transport management is the replacement of rolling stock with more modern vehicles. Provisions of this type are executed at the community, domestic and regional level. There is strong pressure on withdrawing rolling stock with combustion engines. | Increased investment expenditure for rolling stock and infrastructure |
| Increased number of public transport passengers in big cities | The analyses of the number of passengers indicate that the participation of public transport in fulfilling the transport needs of inhabitants increases in huge Polish agglomerations (e.g., Warsaw, Kraków). | Increased revenues from the sale of tickets (applies to large Polish agglomerations). |

Source: own study based on: [10–19].

The tendencies presented in the table are applicable for Poland, but also very similar to the other central and southern European countries [20–23]. The tendencies of negative impacts on the public transport financing systems dominate over the positive ones. Consequently, it should be concluded that the already strained financing system will require an even greater involvement of public funds or significant limitation of the transport offer in the future. Therefore, the public party responsible for public transport management must allocate the funds for investment and maintenance of public transport system as effectively as possible.

*1.2. Public Transport Organisation Models*

In Polish legislation, there are two basic entities which constitute the public transport market: the organiser, and the operator. The organiser of public transport is either a territorial self-government unit (e.g., commune, district, voivodeship), or an association of such units. There may also be a situation in which one or several units will assign the duty of transport organisation to another entity. Such a solution is particularly popular in the case of monocentric urban systems, where economic, educational, administrative and cultural functions are concentrated in one (core) city, whereas other smaller locations mainly have residential functions. The public transport organiser's duties include conducting marketing research to identify the inhabitants' transport needs, preparing transport offers (determination of line course, locations of stops, timetables etc.), contracting transport services, conducting tariff policy and ticket distribution, providing passenger information, promoting public transport, and implementing control functions. On the other hand, the operator is responsible for technical matters related to transport implementation, such as providing the appropriate rolling stock, and employing drivers, as well as operating the line in accordance with the contract and timetable. In settlements between the organiser and the operator, the mode of selection and the related contract type are significant.

The public party represented by the organiser is the party which orders public transport services. The contractor (operator) can be selected through competition proceedings in accordance with the specific public procurement regulations, or in non-competition proceedings. The subject of tender procedure for line operation can be either the entire market, specific parts (packages) of lines, or the operation of individual lines. In tender procedures, the bidders declare the amount for which they will perform operation work in line with the specification. The settlement between the operator (operators) and organiser is made on the basis of the volume of kilometres per month. The amount of payment is the product of rate per kilometre (in accordance with the tender results) and the volume of completed operation work. In this model, the technical aspect of the order implementation is the only risk on the part of the operator. It is mainly related to the necessity to provide the required rolling stock, sufficient number of drivers, and operation of service required by the organiser (e.g., concerning the passenger information). The operator's remuneration does not depend on the number of passengers or level of income from the sale of tickets. The commercial risk in this model is incurred by the organiser, who is responsible for the tariff policy, distribution of tickets, and promotion of provided services among the inhabitants.

An alternative to tender procedure is to entrust transport service to the so-called internal entity based on the provisions of Regulation (EC) 1370/2007 or procurement directives 2014/24/EU and 2014/25/EU. The function of internal entity can only be performed by an enterprise which is 100% owned by a public transport organiser. Therefore, such an entity is dependent on the decision of the organiser, which has a direct impact on the strategic decisions of the entity and controls its actions. Direct ownership and capital connections between the organiser and the operator facilitate cooperation, e.g., in the field of transferring assets (which has been described more extensively in the subsequent subchapter). In the case of establishing an internal entity, the operator is paid compensation, which is a form of public support granted in order to ensure the appropriate standard, intensity, and reach of transport service implementation. However, compensation cannot be granted

in an excessive amount, since in such case it would be regarded as prohibited public aid. The compensation amount cannot exceed the financial result calculated as:

- Costs incurred due to service provision;
- Minus all positive financial inflows generated as a result of providing public services;
- Minus tariff and related revenue;
- Reasonable profit.

In the model with an internal entity, the division of risks between the organiser and the operator depends on the legal grounds of the concluded contract. In the case of referring to Regulation no. 1370/2007, the contract is licence-based. In such a case, the commercial risk is incurred by the operator, which is responsible for establishing the tariff policy (in consultation with the organiser) and the distribution of tickets. Revenue from the sale of tickets goes directly to the operator's budget. The obtained revenue from the sale of tickets, together with the determined compensation amount, must cover the costs of the operator's activity, and provide them with the so-called reasonable profit. On the other hand, in the case of referring to the procurement directives, the operator only incurs the technical risk.

In Poland, in the group of cities with over 50,000 inhabitants, the model based on the dominating share of internal entity is most frequently applied (data for 2018) [24]. In such cases, competition proceedings either do not occur at all, or they are limited to the level of approximately 10–15% of the market, and usually concern the operation of night lines or connections on the outskirts of a city or agglomeration. Only in the case of Upper Silesian cities associated in Municipal Transport Union of the Upper Silesian Industrial District (KZK GOP) and Intermunicipal Public Transport Association (MZKP) in Tarnowskie Góry, the model based on public procurement was fully applied. Currently, the above-indicated organisers together with the Municipal Transport Authority in Tychy have formed the Metropolitan Transport Authority, which is the organiser of public transport in Metropolis GZM.

Generally, the model based on an internal entity, whose organiser is the owner of all or a majority of shares, is a very flexible solution. Because public procurement is not applied in this case, the concluded contracts can be modified depending on current needs. Both the parameters of provided services and their scope can be changed. On the other hand, the organiser's dependence on one entity is a significant risk. The risk also concerns the increasing costs of such an operator's activity, since it is not subject to market price verification.

In the case of the competition-based model, the basic tool is public procurement law, which results from the high value of transport contracts. The application of tender procedures is related to the necessity to plan contracts, which are frequently 8–10 years long. Long-term cooperation must be planned at the stage of preparing the proceedings, providing certain flexibility on the one hand, and securing the appropriate level of public service performance on the other hand. In consequence, proceedings of this type are complex and time-consuming. Moreover, they require hiring qualified employees. The comparison of advantages and disadvantages of both models is provided in the Table 2.

*1.3. Rolling Stock and Infrastructure Charging Management Models*

Regardless of the adopted market organisation model, the technical risk is on the part of the transport operator. For the purpose of implementing transport services, the operator should have specific resources, in particular, the rolling stock and the necessary infrastructure. The practice shows that there are three general methods for organising rolling stock and infrastructure:

- purchase;
- leasing;
- loan (for a consideration) or lending for use (free of charge) the property owned by the transport organiser.

**Table 2.** Strengths and weaknesses of selected public transport market models.

|  | **Competition-Based Model** | **Internal Entity-Based Model** |
|---|---|---|
| Strengths | Market price verification Possibility to change an unreliable operator Competition between operators in the field of price and quality of services | High flexibility of cooperation, e.g., in the field of changing the volume of operation work Possibility to shape the operator's duties freely and achieve the economies of scale Possibility to recapitalise the operator or provide them with assets for use |
| Weaknesses | Submitted public procurement procedures Low flexibility in terms of changing contract terms and conditions | Limited possibilities of cost verification Risk of growing compensation Risk of becoming dependent on an internal entity in the long run |

Source: own study based on [25].

Purchase is the most common form of acquiring rolling stock, especially in the case of implementing long-term contracts and a high volume of operation work. Leasing is a less popular solution, applied in the case of smaller duties or as a substitute of loan for rolling stock purchase. In the case of the Polish public transport market, the least frequently used solution is when the organiser is the owner of the rolling stock and loans or lends it to the operator. The loan or lending procedure is additionally significantly affected by the adopted model of market organisation. In the case of an internal entity, such a solution is relatively simple to introduce. On the other hand, in the case of a model based on competition, a potential loan will be subject to tender procedure.

The implementation of modern and ecological solutions in the field of drives used in public transport is also related to the necessity to provide the relevant infrastructure. This problem was irrelevant in the situation of using a conventional drive due to the prevalence of owned (company) or chain fuel stations. Taking into consideration alternative drives, the above issue becomes more complicated. Battery-powered buses (EV) require the appropriate charging infrastructure, including plug-in chargers and pantograph chargers. Here appears a certain duality, resulting from the fact that plug-in chargers will be used for long charging, most frequently in the depot at night, whereas pantograph chargers will be used for quick charging during standstill, e.g., node stops, or end stops. This has a significant impact on the infrastructure solutions [26].

In the case of hydrogen fuel-cell buses, the key infrastructure element is a hydrogen filling station. In this case, the most frequent solutions consist in delivering and installing filling stations by the hydrogen supplier.

Apart from the above technical factors, the rolling stock and infrastructure management model will also be affected by the possibility to obtain external funding from public sources. The practice shows that both the organiser and the operator may apply for funding. On the other hand, certain financial support instruments are intended only for the organisers.

*1.4. Goal of the Article*

The goal of this article is to analyse the effectiveness of financial investment in the public transport rolling stock. The following three types of drives were analysed: conventional (diesel), battery-powered (EV), and hydrogen fuel-cell (FCEV). Additionally, different forms of rolling stock and infrastructure possession were taken into consideration. The main research questions with reference to the presented one are as follows:

- what is the level of financial effectiveness of particular bus drive technologies?
- what is the impact of rolling stock and infrastructure management model on investment decisions?
- how does high public subsidy affect the investment performance indicators?

The response to these questions will be based on simulations carried out for a large investment project implemented in the biggest metropolitan area in Poland (Metropolis GZM). The project consisted in the purchase of 32 buses, together with infrastructure, by the transport organiser.

## 2. Materials and Methods

### 2.1. Characteristic of the Study Area

In order to proceed with the financial analysis, there was used an example of an investment project executed by the Metropolis GZM, which is the organizer of the public transport on the metropolitan area associating 41 municipalities located in the Silesian voivodeship (southern Poland). This area is characterized by high urbanization level and is one of the biggest conurbations in central Europe (27 thousand sq. km). The potential of this area is based on 14 core neighbouring large cities surrounded by smaller neighbourhoods. The population of GZM in 2020 counted 2.2 million inhabitants. The area is responsible of delivery of ca. 8% Polish GDP. Dense agglomeration of cities, the area, and the diversity of municipal services create big challenges for the management of delivery of public transportation services to the community.

The public transport service in Metropolis GZM is provided by the Metropolitan Transport Authority. This unit is the organiser of services, which means that in accordance with Art. 8 of the Act of 16 December 2010 on public transport, its duties consist in transport development planning, public transport organisation and public transport management. The detailed scope of organiser's duties has been specified respectively in Art. 9, 15, and 43 of the above-mentioned Act. The carriers (operators) in the metropolis area were selected in a public procurement procedure, and they are entitled to remuneration for the transport services provided (order processing fee). The municipal bus lines are served by 39 operators, whereas the tram and trolleybus lines are served by one entity each.

### 2.2. Methods and Data

For the purpose of this study, a number of scenarios for the functioning of public transport system were specified, based on the example of the bus carrier, which intends to purchase a new rolling stock within the framework of a project. Due to the requirements specified in Art. 68(4) of the Act of 11 January 2018 on electromobility and alternative fuels, public transport operators must provide an appropriate number of low-carbon or zero emission buses within the deadline specified in the legal regulations. Therefore, it was assumed that 32 battery-powered buses (EV) or hydrogen fuel-cell (FCEV) buses will be introduced in the metropolis areas and replace the previously used vehicles. The legislator allows for a situation in which a public transport operator provides services using the fleet delivered by the organiser. This was the reason for adopting the scenarios in which FNPV indices were calculated. In each variant, it was assumed that operation work performed with the new rolling stock will be identical to the withdrawn rolling stock—the replacement will amount to 100%. In the case of scenarios involving European Union funding, it was adopted that the funding amounts to 85% of net investment expenditure. For the variants with own funding, FNPV values, including the full mix of funding and the involvement of own capital, are the same. Depreciation was omitted each time due to its cost nature (rather than the expense nature). The scenarios are presented in the Table 3. Other model parameters remain unchanged.

Public services are delivered by entities which operate reasonably, taking decisions on the basis of the balance of costs and benefits. Each decision taken by a public entity is reflected in the use of limited resources, and is intended to provide social benefits, which may be indirect, direct, material, or non-material, and therefore, the possibility to determine their value may be different. In the case of transport projects, decisions are taken on the basis of a set of effectiveness assessments, including financial effectiveness, mainly based on possible sources of project financing and social-economic effectiveness, taking into consideration the social-economic impact of the project.

**Table 3.** Characteristics of variants.

| Scenario | Owner | Fuel/Drive | Financing |
|----------|-------|------------|-----------|
| A | Operator | EV | Own |
| B | Operator | EV | Own + EU |
| C | Operator | FCEV | Own |
| D | Operator | FCEV | Own + EU |
| E | Operator | Diesel Euro 6 | Own |
| F | Operator | Diesel Euro 6 | Own + EU |
| G | Organiser | EV | Own |
| H | Organiser | EV | Own + EU |
| I | Organiser | FCEV | Own |
| J | Organiser | FCEV | Own + EU |
| K | Organiser | Diesel Euro 6 | Own |
| L | Organiser | Diesel Euro 6 | Own + EU |

Source: own study.

If a reasonable entity has a choice, it strives to maximise their benefits in the nearest future, since the potential future benefits carry the risk of failure to obtain, e.g., due to a change of the project implementation conditions. The concept of money value change over time takes into account the fact that the money spent or obtained in future periods will have a different value than the money spent or obtained at present [27]. To be capable of assessing the general cumulated effect of project implementation, it is necessary to determine the future stream of costs and benefits, and then to reduce the future flows to the current value. For this purpose, the net present value (NPV) discount method is universally applied. A positive result of NPV evidences the fact that the project generates a surplus of profits over costs in the future, whereas in the opposite situation, it generates deficit. In the classic approach to the application of NPV method to assess the effectiveness of projects in a particular organisation, only revenue and accounting costs are adopted for calculations, together with cash flow, without taking into consideration non-material benefits and costs not quantifiable in currency, indirectly, which are identifiable in the project surroundings. The NPV result should also take into consideration the residual value of the project. Next, the internal rate of return (IRR), i.e., the discount rate for which the NPV value amounts to zero, is determined for a particular flow stream.

An interesting approach toward efficiency analysis is proposed by Wąsowicz [28]. He proposes a differentiation between strategic and operational efficiency. When analysing operational efficiency, various techniques can be used, but in practice, the most common are the relationships—dependencies between quantities, coefficients and indicators. Indicators, i.e., relative values, and ratios are informative measures of various types of activity. From the financial perspective, Wąsowicz promotes application of financial indicators of returns, liquidity, and debt. These values are particularly useful for short term decision making for operations management of the carrier, but due to their nature only partially take into account the overall cash outflow and inflow, and net cash flows for the investment projects in the public transport.

As results from the research conducted by Annema et al. [29] demonstrate, the indirect and external effects of project implementation are not always taken into consideration in the assessment of effectiveness, or they are included in a qualitative manner. Therefore, in order to make a reliable assessment of project effectiveness in the financial aspect, it is necessary to use direct and measurable data on the costs and benefits.

In the case of projects which have the possibility to obtain EU funding, the assessment of financial effectiveness must be performed from at least two perspectives, depending on the funding source. In the first case, the value of financial NPV marked with FNPV/C

symbol is determined for the entire project, taking into consideration only the operating flows and the investment expenditure which it generates. The second financial parameter that should be determined for the project is its updated net value, taking into consideration the domestic funding marked with FNPV/K symbol. In this case, cash flows take into account the potential revenue that may come from the EU funds. In the situation when a project is carried out by a private entity and obtains support from domestic public funds, it is also necessary to make the assessment of financial effectiveness, taking into consideration only the private capital. In such a situation, the revenues from domestic public funds are also taken into consideration in the cash flow statement. The updated net value is then marked with FNPV/Kp symbol. Similarly, the appropriate financial return rates (FRR) are determined for each FNPV value.

The traditional approach to NPV calculation treats future cash flows as firm (deterministic) values. However, many studies on NPV estimation are based on the assumption that these data are uncertain, which is entirely justified, since both revenues and expenses related to the project concern the future (with the exception of the cash flows at moment 0). There are several different methods for including uncertainties in such calculations, for example: (1) increase of discount rate, (2) application of sensitivity analysis, (3) comparison of pessimistic and optimistic cash flow scenarios, and (4) estimation of expected cash flows by means of scenario planning and probability distribution [30]. Certain estimations can be determined with a significant degree of accuracy, but others can only be determined to a relatively broad extent. When the uncertainty level is relatively high, the extent of estimation is relatively broad, and NPV can only be quantified with limited certainty. To assess the changeability of NPV level depending on the selected variables, the sensitivity analysis is universally applied. Its task is to identify the critical variables, whose increase or decrease will cause a significant change of NPV value [31]. Therefore, the sensitivity analysis allows for the checking of how NPV changes, depending on the level of components of specific cash flows. Moreover, it also provides the possibility to determine the range for cash flows at a particular moment, in which NPV remains positive or higher from the NPV of an alternative project. Therefore, the sensitivity analysis is intended to identify the critical variables of the project. It is performed by changing the selected variable for the project by a specific percentage value (ceteris paribus), and observation of the resulting changes in selected indices [32]. For European projects, critical variables are ones whose change by 1% causes a change of NPV value by at least 1%.

In the literature on the subject, we may find different points of view on the discount rate which should be adopted for the financial analysis of investment projects. According to a general and relatively uncontroversial definition, financial discount rate is an alternative cost of capital. Alternative cost means that when capital is used in one project, there is no possibility to receive return in another project. Therefore, when we involve capital in a single investment project, we lose the possibility to obtain benefits from an alternative project.

There are basically three approaches which may be helpful in determining the relevant financial discount rate. In the first approach, the minimum alternative cost of capital is estimated. In this approach, the real discount rate should measure the cost of capital used in a particular investment project. In consequence, the reference point for a public project could be the real return rate from government bonds (marginal cost of public debt) or long-term real interest rate of commercial loans (if the project requires private funding). If the use of many sources of financing is assumed, then the discount rate should be determined by establishing the weighted average cost of capital (WACC).

This approach is very simple, but it can produce incorrect results. It should be remembered that, in this approach, we use the actual cost of capital to determine an alternative cost of capital, whereby these two notions are different. In reality, the best alternative project may bring significantly more benefits than the actual interest rate of public or private loans amounts to.

The second approach determines the maximum border value of discount rate, since it takes into account the lost profit from the best investment alternative. In practice, the alternative cost of capital is estimated on the basis of the marginal rate of return from the securities' portfolio in the international financial market, in the long run and with minimum risk. In other words, the alternative to project revenue is not public or private debt buyout, but a return from an appropriate investment portfolio.

The third approach consists in determining the border rate. This means the withdrawal from a detailed examination of the specific cost of capital for a particular project (in the first approach), or from the analysis of specific portfolios in international financial markets or alternative projects for a particular investor (in the second approach), and the use of a simple rule.

For the projects in the financial perspective of 2014–2020, the discount rate was specified at the level of 4%. It was determined on the basis of the average return from investment in small, medium-sized, and large stock exchange companies, international companies, bonds and cash investments.

## 3. Results

### 3.1. Analysis of Investment Expenditure

Depending on the adopted variant, investment expenditure is different—regardless of whether the investor is the organiser or the operator, they are the same. The summary of net unit expenditure was presented in the Table 4.

**Table 4.** Assumptions concerning unit investment expenditure.

| Variant | Bus 12 m [mln EUR] | Bus 18 m [mln EUR] | Charging/Fuelling Infrastructure [mln EUR] |
|---|---|---|---|
| Diesel Euro 6 | 0.22 | 0.29 | Not applicable |
| FCEV | 1.0 | 1.15 | Not applicable |
| EV | 0.45 | 0.55 | 0.05 |
| Diesel Euro 6 | 0.22 | 0.29 | Not applicable |

Source: own study.

In the case of EV vehicles, it was necessary to adopt replacement expenditure after seven years of operation, due to the lifespan of traction batteries.

27 Maxi class buses (12 m) and 5 Mega class buses (18 m) were adopted for the analysis. The EV variant also assumes the purchase of plug-in depot chargers.

The use of the changed drive in vehicles will not affect the number of transported passengers.

The data concerning the amount of expenditure have been determined on the basis of information obtained from public tenders for deliveries of rolling stock and charging infrastructure in 2021.

### 3.2. Analysis of Operating Costs

Depending on the adopted organisational model, different methods are used to calculate the costs of operating activities, whose precise determination is necessary in order to establish the FNPV/C and FNPV/K values using the differential method. The analysis was performed on the basis of financial documents of the organiser and operators, as well as interviews with the accounting services of these organisations.

In the case of the organiser of the service being the owner of the fleet, the costs they incur for implementation of the investment are entirely classified as outsourced services. Both the activities related to vehicle maintenance and the costs related to performing operation work by operators (carriers) are charged as the same type of costs. The analysis of other costs by type demonstrated that they are independent from implementation of the investment. To sum up, in all variants in which the organiser is the owner of the fleet and infrastructure, there are 2 groups of costs of outsourced services: lump sum costs of

vehicle maintenance, and costs of service provision, depending on the amount of completed operation work. In accordance with the concluded agreements between the organiser and the carriers, the basis for their remuneration is the rate per 1 vehicle-kilometre, which is subject to annual valuation.

In the case of the operator (carrier) being the owner of the fleet, it is also possible to indicate the costs dependent on implementation of the investment. These may particularly include costs by type: consumption of materials and energy, and outsourced services. The diversity of costs of materials and energy mainly results from technical aspects, since in different variants, vehicles must be provided with fuel measured in different units or directly with electricity, whose consumption is expressed directly in kWh. In the case of traditional drive, it is necessary to include the part relating to traction and the part emitted in the form of heat in the balance of energy delivered to the vehicle. Therefore, it is necessary to take into account the general energy efficiency of such a drive system. The analysis of costs by type, such as remuneration, social insurance and employee benefits, taxes, and fees, as well as outsourced services, demonstrated that they are independent from the investment.

In accordance with the NPV calculation method, depreciation, which is not an expense, and is not included in the differential cash flow balance, has been omitted.

### 3.3. Comparison of the Effectiveness of Variants

The analysis of cost effectiveness was performed for the above-described variants of investment implementation using the NPV method. The discount rate of 4%, recommended by EU institutions and domestic financing institutions, was adopted for the calculation. The results of calculations were included in the Table 5. A 10-year time horizon was assumed in order to calculate the FNPV Value, which results from the 10-year vehicle depreciation period, regardless of the ownership formula.

**Table 5.** Results of effectiveness analysis for particular variants.

| Scenario | Fuel/Drive | FNPV/C [mln EUR] | FNPV/K [mln EUR] |
|----------|------------|------------------|------------------|
| A | EV | −13.233 | Not applicable |
| B | EV | −13.233 | 0.051 |
| C | FCEV | −27.607 | Not applicable |
| D | FCEV | −27.607 | 0.231 |
| E | Diesel Euro 6 | −6.245 | Not applicable |
| F | Diesel Euro 6 | −6.245 | −0.085 |
| G | EV | −17.013 | Not applicable |
| H | EV | −17.013 | −3.667 |
| I | FCEV | −54.078 | Not applicable |
| J | FCEV | −54.078 | −27.312 |
| K | Diesel Euro 6 | −11.461 | Not applicable |
| L | Diesel Euro 6 | −11.461 | −4.881 |

Source: own study.

All FNPV values are negative, which demonstrates that public transport services have no potential to generate profit; however, the most favourable from the financial perspective is variant E or F. From the point of view of the operator, FNPV/K should oscillate around 0, as the project is not aimed at generation of excessive profit. All of the operators functioning in the GZM area are limited liability companies, so they need to generate profit, but due to their capital structure and mission, it can be close to zero. For scenarios G through L, where the organiser is responsible for delivery of the vehicles, the overall FNPV/K may be negative as, in such a case, one is obliged to cover the deficit on this particular public service either from other sources or by increase of debt. The organiser is a public

legal person–metropolitan government unit. For all scenarios, a sensitivity test was also performed with single variable–capital expenditure. The results are presented below in the Table 6.

**Table 6.** Sensitivity analysis for FNPV/C, FNPV/K and relative changes of CAPEX.

| Scenario | Fuel/Drive | CAPEX Change [%] | FNPV/C Change [%] | FNPV/K Change [%] |
|----------|-----------|------------------|-------------------|-------------------|
| A | EV | ±1% | ±1.17 | Not applicable |
| B | EV | ±1% | ±1.17 | ±84.81 |
| C | FCEV | ±1% | ±1.17 | Not applicable |
| D | FCEV | ±1% | ±1.17 | ±11.48 |
| E | Diesel Euro 6 | ±1% | ±1.24 | Not applicable |
| F | Diesel Euro 6 | ±1% | ±1.24 | ±3.28 |
| G | EV | ±1% | ±0.88 | Not applicable |
| H | EV | ±1% | ±0.88 | ±0.61 |
| I | FCEV | ±1% | ±0.58 | Not applicable |
| J | FCEV | ±1% | ±0.58 | ±0.17 |
| K | Diesel Euro 6 | ±1% | ±0.63 | Not applicable |
| L | Diesel Euro 6 | ±1% | ±0.63 | ±0.22 |

Source: own study.

## 4. Discussion and Conclusions

The purpose of the paper was to assess financial effectiveness of particular bus drive technologies and models of delivery of classic mobility services in the city. Through analysing various scenarios and models of financing and ownership of the fleet, it was demonstrated that the use of traditional, diesel fuel-powered combustion engines provides the highest financial effectiveness regardless of the model of ownership. Due to the differences of unit prices per vehicle depending on the drive applied, the most traditional one is still the least expensive and offers the best value for money.

Beside the drive, the most important difference in the values of indicators depend on:

- availability and application of external financing;
- model of ownership of the fleet.

In each case, an additional stream of financing from the EU increases FNPV/K values; however, in the case of the scenarios of the organiser's ownership of the vehicles, costs such as depreciation and amortisation or technical maintenance must be covered on their own. Moreover, due to delivery of the services by the operators using the fleet delivered by the organiser, the cost of materials and energy, salaries, and social security charges, increased by the amount of sound (just) profit, must be paid to the operators in the form of compensation. It additionally lowers values of both FNPVs. Moreover, execution of the project does not affect the levels of sales revenues, as the services must be delivered for the public, practically regardless of the financial conditions. If there is a loss generated on delivery of such services, regardless of the financing mix of the investment, the deficit must be covered by the organiser from other sources or as an increase of debt.

In the case of ownership of the fleet by the operators, the FNPVs are higher. Naturally, they incur all of the operating cash costs of delivery of the services, and they also incur depreciation and amortisation, but in comparison to the organiser, they are entitled to receive higher compensation according to provisions of Regulation (EC) 1370/2007. Profits generated on delivery of public transportation services should not be excessive and taking into account external, public financing of investment (e.g., EU grant), the overall effectiveness of the project should be close to 0. In case of loss on delivery of the services, the operator is then obliged to cover it from other sources. Nevertheless, flexibility of

making up the loss on particular services is higher, as the remuneration for delivery of public services is subject to annual audit and, in the case of huge deficit, the operator may always apply for an increase of the unit rate per km.

Public subsidies have tremendous impact on the performance indicators of the projects. This applies to all scenarios. In the case of a successful application for funding, in the years 2014–2020, it was possible to receive a grant covering 85% of eligible expenditure. From the accountancy perspective, not all of the expenditure, but only a part covered by investor's contribution, was subject to deduction as depreciation and amortisation, so it also affected the profitability of the operator, as one did not need to charge the financial results with 100% cost of fleet depreciation and amortisation. On the other hand, it also affected the possibility of recovery of cash, as only a part of this cost could be taken into account in the cash flow statement.

Due to the diversity of financing sources available both for organisers and operators, in recent years there has been also a political discussion concerning the best available solutions in terms of satisfying passengers' needs and improvement of quality of mobility services. The main challenge in the approach presented in all scenarios is the choice of the investor and owner of the fleet. In the case of delivery of the vehicles to the operators, all FNPV calculations refer to solely these organisations, thus they do not reflect the efficiency of the overall delivery of transportation services to the public. They only present the results from the point of view of the company. In the case of delivery of the vehicles to the organiser, FNPV calculations take into account all potential revenues, costs, and expenditures related to functioning of the public transportation system. Therefore, in the second case, the presented estimations are closer to real costs of delivery of this public service that must be covered by the metropolitan government. In the first case, all efficiency estimations refer to a single company only, thus they reflect the value of the project only for them, and oversee the real efficiency of delivery of public transportation service on given area.

One of the fundamental factors affecting the environmental impacts, but also financial efficiency of urban public transport is the source of energy. According to the statistics presented in [33], in Poland—compared to other EU Member States—still dominates production of electricity from solid fuels incineration. The share of wind energy is comparable to the EU average, however the share of biogas, biomass, solar, and water energy is significantly lower than the average. For instance, the share of solar energy or heat pumps in Poland is roughly seven times lower in the EU on average. Based on Regulation (EU) 2018/842, the reduction target for Poland for greenhouse gas emissions in non-ETS sectors (not covered by the European Union Emission Trading Scheme) has been set at −7% in 2030, compared to the 2005 level. Development of fuel cell electric vehicles has a significant impact on the decarbonisation of transport, which currently relies heavily on diesel and petrol. At present, e-mobility significantly reduces transport emissions, but if one wishes to decarbonise the transport sector completely, it is crucial to consider wider implementation of zero-emission vehicles, including those powered by hydrogen. It is worth stressing that the potential for using hydrogen should not be sought only in car transport, but also in rail, aviation, and maritime applications [34]. The forecasts for the future, according to the National Energy and Climate Plan 2021–2030, assume linear increments of the share of renewable energy sources (RES) to reach roughly 23% in year 2030.

**Author Contributions:** Conceptualization, T.W.S. and G.K.; methodology, T.W.S. and G.K., validation, T.W.S., G.K. and S.T.; formal analysis, T.W.S.; resources, T.W.S., G.K. and S.T.; data curation, T.W.S. and G.K.; writing—original draft preparation, T.W.S., G.K. and S.T.; writing—review and editing, T.W.S. and G.K.; supervision, T.W.S.; project administration, G.K.; All authors have read and agreed to the published version of the manuscript.

**Funding:** This research received no external funding.

**Institutional Review Board Statement:** Not applicable.

**Informed Consent Statement:** Not applicable.

**Data Availability Statement:** Not applicable.

**Conflicts of Interest:** The authors declare no conflict of interest.

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
