# Peer review of "Models of Delivery of Sustainable Public Transportation Services in Metropolitan Areas–Comparison of Conventional, Battery Powered and Hydrogen Fuel-Cell Drives"

_energies, doi:10.3390/en14227725_

Round 1
Reviewer 1 Report
1) Abstract should report more quantitative information
2) Lines 248-372: this method description should be shortened: some paragraphs can be removed or moved in the introduction section.
3) The discussion should mention the actual share of renewable energy used for electricity production in Poland, and a realistic forecast (possibly supported by official National reports) on how much it will increase in the next decades.
Author Response
Thank you for a very constructive review. Following your kind remarks, we have introduced the following amendments:
- quantitative paramteres were added to the introduction, however it was not possible to the values of the analysis.
- the method was reviewed and modified slightly, but the core text was left without major changes as it would be problematic to follow our reasoning and logic if we removed a part of that. Anyway thank you for this remark, we will bear it in mind for our next works.
- In the discussion there was made reference to the article concerning comparison of energy mix in Poland vs EU as well as to the National Energy and Climate Plan 2021-2030 which includes the official forecasts of transformation into low-emission sources.
Reviewer 2 Report
The article is well documented and clear presented. The methodology is explained in detail, documented. Although, at the line 67, a reference is needed in order to support the affirmation "The tendencies presented in the table ... but also very similar to the other Central and Southern European countries."
Author Response
Thank you for your kind remarks.
As far as your suggestion is concerned, we added necessary references to the articles concerning aging and other challenges for public transportation services in terms of their power and delivery of energy.
References no. 20-23.
Reviewer 3 Report
The article is devoted to current and important issues paying attention to the relationship between applied technologies in urban transport and financial efficiency. The article carries out a consideration in the field of possibilities of application of three technologies of bus propulsion and models of providing classic urban mobility services in the context of economic factors. It should be noted that the high costs of functioning of the market mechanism through a specific price system sometimes justify the public provision of goods for which
marginal cost exists. Thus, the cost of the services offered must be structured in such a way as to be socially acceptable. This means that the constituting authority is obliged to subsidise the activities of a municipal public utility, the activities of which may not be profitable, but are necessary due to the need to satisfy the needs of the local population.
The article has an adequate theoretical basis, relevant information and analysis, good partial (in the article) and final (in the conclusion) conclusions. The article uses original research by the author and cited research by other researchers, which enriches its content. It is written in good language and is based on an analysis of current and well-chosen literature. Thus, the article should be regarded rather as an interesting introduction to a very important issue and treated as a scientific article.
The reviewer lacked a reference to profitability ratios, which inform about the effectiveness of activity and express the relation of profit calculated on different levels of economic activity to: achieved revenue from sales of products, assets (assets), equity, employment. In this regard, I recommend the authors the book by K. Wąsowicz, Efektywność przedsiębiorstw użyteczności publicznej lokalnym transportu zbiorowego, Kraków 2018.
Author Response
Thank you for your kind remarks.
We analyzed the proposed reference by Wąsowicz K. and the methods of efficiency evaluation were included in the methodological part of the article.
Appropriate reference was included in the bibliography of the paper.